# Impact of Observational Time Window on Coupled Data Assimilation: Simulation with a Simple Climate Model

Yuxin Zhao[1], Xiong Deng[1,2], Shaoqing Zhang*[3], Zhengyu Liu[4,5], Chang Liu[1,2], Guijun Han[6], Xinrong Wu[6]

[1]College of Automation, Harbin Engineering University, Harbin, 150001, China
[2]GFDL-Wisconsin Joint Visiting Program, Princeton, NJ08540, USA
[3]Physical Oceanography Laboratory/CIMST, Ocean University of China and Qingdao National Laboratory for Marine Science and Technology, Qingdao, 266100, China
[4]Atmospheric Science Program, Department of Geography, Ohio State University, Columbus, Ohio, 43210, USA
[5]Laboratory for Climate and Ocean-Atmosphere Studies (LaCOAS), Department of Atmospheric and Oceanic Sciences, School of Physics, Peking University, Beijing, 100871, China
[6]National Marine Data and Information Service, Tianjin, 300171, China

*Correspondence to: Shaoqing Zhang (szhang@ouc.edu.cn)

**Abstract.** Climate signals are the results of interactions of multiple time scale media such as the atmosphere and ocean in the coupled earth system. Coupled data assimilation (CDA) pursues balanced and coherent climate analysis and prediction initialization by incorporating observations from multiple media into a coupled model. In practice, an observational time window (OTW) is usually used to collect measured data for an assimilation cycle to increase observational samples that are sequentially assimilated with their original error scales. Given different time scales of characteristic variability in different media, what are the optimal OTWs for the coupled media so that climate signals can be most accurately recovered by CDA? With a simple coupled model that simulates typical scale interactions in the climate system and "twin" CDA experiments, we address this issue here. Results show that in each coupled medium, an optimal OTW can provide maximal observational information that best fits characteristic variability of the medium during the data blending process. Maintaining correct scale interactions, the resulting CDA improves the analysis of climate signals greatly. This simple model results provide a guideline when the real observations are assimilated into a coupled general circulation model for improving climate analysis and prediction initialization by accurately recovering important characteristic variability such as sub-diurnal in the atmosphere and diurnal in the ocean.

## 1   Introduction

Currently, the interactions between the earth climate system's major components, such as the atmosphere, ocean, land and sea ice, have been reasonably simulated by coupled climate models, which can also give the evaluation of climate changes (Randall et al., 2007). However, because of the uncertainties and errors in models (e.g., parameterization is only an approximation to sub-grid processes and dynamical core is imperfect), models always tend to produce different climate features and variability from the real world (e.g. Delworth et al., 2006; Collins et al., 2006; Zhang et al., 2014). Due to the significant importance of preserving the balance and coherence of different model components (or media) during the coupled

model initialization, data assimilation for state estimation and prediction initialization should be performed within a coupled climate model framework (e.g. Chen et al., 1995; Zhang et al., 2007; Chen, 2010; Han et al., 2013). The characteristic variability time scales of different media within the coupled frameworks are usually different. When the observed data included in one or more components of the coupled system framework are assimilated, the observational information will be able to be transferred among different media through the coupled dynamics so that all media gain consistent and coherent adjustments. Such an assimilation procedure is called coupled data assimilation (CDA), which can sustain the nature of multiple time-scale interactions during climate estimation and prediction initialization (e.g. Zhang et al., 2007; Sugiura et al., 2008; Singleton, 2011), thus producing better climate analysis and prediction initialization and therefore improving the coupled models' predictability (Yang et al., 2013). Zhang et al. (2007) developed the first CDA system in a fully coupled general circulation model, the version 2 of Geophysical Fluid Dynamics Laboratory Coupled Model (GFDL CM2). The National Centres for Environmental Prediction (NCEP) also started using coupled models to generate first-guess forecasts for their Climate Forecast System Reanalysis (CFSR, Saha et al., 2010). Despite the enormous benefits and demand for CDA, it remains both theoretically and technically challenging to implement strong CDA in fully-coupled models, including the estimation of the coupled model error covariance matrix and the huge computational costs (e.g. Han et al., 2013; Lu et al., 2015; Liu et al., 2016).

During the coupled data assimilation process, usually an observational time window (OTW) is used to collect measured data in each medium for an assimilation cycle (e.g. Pires et al., 1996; Hunt et al., 2004; Houtekamer and Mitchell, 2005; Laroche et al., 2007) to increase observational samples. As in Hunt et al. (2004), we expand the EnKF to include a time window in which the observations are treated as the exact assimilation times, even though their times are different in the window. Namely, we just assume that all the collected data sample the "truth" variation at the assimilation time and will be sequentially assimilated with their original error scales. Thus the OTW is applied in a 3-dimensional data assimilation fashion rather than a 4-dimensional one. Apparently, while a large OTW provides more observational samples at the assimilation time, the assimilation process blends more data from different times and may distort variability being retrieved. Given the fact that climate signals are the results of interactions of multiple time scale media, correct variability retrieved for each medium so as correct scale interaction maintained in CDA is particularly important for climate analysis and prediction initialization. In this study we attempt to answer the following two questions: 1) What is the impact of varying OTWs for each coupled component within the coupled model framework on the quality of CDA? 2)Based on this impact, does an optimal OTW exist so that assimilation fitting has maximum observational information but minimum variability distortion?

With a simple conceptual coupled climate model and a sequential implementation of the ensemble Kalman filter, this study first analyses the characteristic variability time scale of each coupled medium and identifies the corresponding optimal OTW. Then the impact of optimal OTW on the quality of CDA and its linkage with the corresponding time scale of characteristic variability are investigated. The simple coupled model consists of three typical components, including the synoptic atmosphere (Lorenz, 1963) and the seasonal-interannual slab upper ocean (Zhang et al., 2012) coupling with the decadal deep ocean (Zhang 2011a,b). Although the simple conceptual coupled model does not share the similar complex physics with

a coupled general circulation model (CGCM), it does reasonably simulate the typical interactions between multiple time-scale components in the coupled climate system (see Zhang et al., 2013). The simple coupled model helps us understand the essence of the problem by revealing the relationship between the optimal OTWs and corresponding time scales of characteristic variability as well as their impact on CDA. The low-cost nature of the simple model also provides convenience

for a large number of CDA experiments with different OTWs in optimal OTW detection. The ensemble Kalman filter (e.g. Evensen, 1994; 2007; Whitaker and Hamill, 2002; Anderson 2001; 2003) used in this study is the ensemble adjustment Kalman filter (EAKF, e.g. Anderson 2001; 2003; Zhang and Anderson, 2003). Using the EAKF with the simple coupled model, we first establish a twin experiment framework. Within such a framework, the degree by which the state estimation based on a certain OTW recovers the truth is an assessment of the influence of the OTW on the quality of CDA. By such a

way, the optimal OTW of each medium is detected and the impact of optimal OTWs on CDA is evaluated. We also discuss the influence of model bias on optimal OTW through biased twin experiment setting.

This paper is organized as follows. Section 2 briefly describes the simple conceptual coupled model, the ensemble adjustment Kalman filter, as well as the twin experiment framework including perfect and biased settings. With a simplest case, we first show the influence of OTWs on assimilation quality and its linkage with the time scale of characteristic

variability in section 3. Then section 4 presents results on detection of the optimal OTWs for different media and the impact of optimal OTWs on CDA. The influence of realistic assimilation scenarios on optimal OTWs is discussed in section 5. Finally, summary and discussions are given in section 6.

## 2   Methodology

### 2.1 The model

Due to the complicate physical processes and huge computational cost involved, it is inconvenient to use a CGCM to investigate the impact of the different OTWs on the analysis of climate signals so as to detect each coupled medium's optimal OTW. Instead, here we employ a simple coupled "climate" model developed by Zhang (2011a). This simple model is based on the Lorenz's 3-variable chaotic model (Lorenz, 1963) that couples with a slab upper ocean (Zhang et al., 2012) and a simple pycnocline predictive model (Gnanadesikan, 1999). Although very simple with low computational cost, in

terms of multi-scale interaction inducing low-frequency climate signals, this model shares fundamental character with a CGCM and it is very suitable for addressing the problem that is concerned here. And for the readers' convenience, here we simply review some key aspects of this conceptual coupled model. With all quantities being given in non-dimensional units, the governing equations are:

$$\dot{\mathcal{X}}_1 = -\sigma\mathcal{X}_1 + \sigma\mathcal{X}_2$$
$$\dot{\mathcal{X}}_2 = -\mathcal{X}_1\mathcal{X}_3 + (1 + \mathcal{C}_1\omega)k\mathcal{X}_1 - \mathcal{X}_2$$
$$\dot{\mathcal{X}}_3 = \mathcal{X}_1\mathcal{X}_2 - b\mathcal{X}_3 \qquad\qquad (1)$$
$$\mathcal{O}_m\dot{\omega} = \mathcal{C}_2\mathcal{X}_2 + \mathcal{C}_3\eta + \mathcal{C}_4\omega\eta - \mathcal{O}_d\omega + \mathcal{S}_m + \mathcal{S}_s\cos(2\pi t/\mathcal{S}_{pd})$$
$$\Gamma\dot{\eta} = \mathcal{C}_5\omega + \mathcal{C}_6\omega\eta - \mathcal{O}_d\eta$$

where $\mathcal{X}_1$, $\mathcal{X}_2$ and $\mathcal{X}_3$ represent the atmospheric model states while $\omega$ and $\eta$ denote those for upper and deep ocean, respectively. A dot above the variable denotes the time tendency. The atmosphere model states are the high frequency variables while the slab oceanic variable $\omega$ is of a lower frequency. For sustaining the chaotic nature of the atmosphere in

reality, the standard values of the parameters included in the atmospheric component ($\sigma$, $k$ and $b$) are set as 9.95, 28 and 8/3, respectively. In the equation of $\omega$, the parameters $\mathcal{O}_d$ and $\mathcal{O}_m$ denote the damping coefficient and heat capacity of the upper slab ocean, respectively. Due to the lower frequency of $\omega$ than that of the model states in the atmospheric components, the time scale of the upper slab ocean variable must be much slower than that of the atmospheric model states. Thus the damping rate parameter ($\mathcal{O}_d$) should be much smaller than the heat capacity, namely, $\mathcal{O}_d \ll \mathcal{O}_m$. Here following Lorenz's idea (Lorenz,

1963), the atmospheric time scale is defined as the typical time by which the atmosphere goes through an attractive lob as one non-dimensional time unit (TU) $\sim O(1)$. We set the parameters $(\mathcal{O}_m, \mathcal{O}_d)$ as (10,1), which show that the slab oceanic variable's time scale is $\sim O(10)$, i.e. 10 times of that of the atmospheric model states. While the $\mathcal{S}_m + \mathcal{S}_s\cos(2\pi t/\mathcal{S}_{pd})$ represents the external forcing, the parameter $\mathcal{S}_{pd}$ denoting the model seasonal cycle is set as 10 to make sure that the period of the external forcing is comparable with the upper slab ocean variables' time scale. In this simple coupled model, the

seasonal cycle is set as 10TUs and thus a model year (decade) equals to 10 (100) TUs. The parameters $\mathcal{S}_s$ and $\mathcal{S}_m$, denoting the magnitudes of the external forcing's seasonal cycle and annual mean, are insensitive to the coupled model and set as (1,10). The coefficients $\mathcal{C}_1$ and $\mathcal{C}_2$ in the equations of $\mathcal{X}_2$ and $\omega$ are used to implement the coupling between the fast atmosphere model states and the upper slab oceanic variable and set as (0.1,1), with that $\mathcal{C}_1$ denotes the upper slab oceanic forcing on the atmosphere while $\mathcal{C}_2$ denotes the atmosphere forcing on the ocean. In addition, $\mathcal{C}_3$ and $\mathcal{C}_4$ represent the deep

oceanic forcing and the nonlinear interaction between the upper and deep ocean. In order to make sure that the atmospheric forcing plays a dominant role in the upper slab ocean, the magnitudes of $\mathcal{C}_3$ and $\mathcal{C}_4$ should be lower than that of $\mathcal{C}_2$ and both set as 0.01. Same as in Zhang (2011a), the deep ocean model state variable $\eta$, denoting the anomaly of pycnocline depth in the deep ocean, is derived from the two-term balance model of the zonal-time mean pycnocline (Gnanadesikan, 1999). Within the equation of $\eta$, the parameter $\Gamma$ is kept constant and the ratio of $\Gamma$ and $\mathcal{O}_d$ denotes the deep ocean variable's time

scale. The time scale of deep ocean variable is longer than that of slab ocean, defined by the relative magnitude of $\Gamma$ to $\mathcal{O}_d$ ($\Gamma$ is set as 100). Similar to the equation of $\omega$, the coefficients $\mathcal{C}_5$ and $\mathcal{C}_6$ denote the linear slab oceanic forcing and the nonlinear interaction between upper and deep ocean. Also for guaranteeing that the linear interaction is dominant and the nonlinear interaction is weaker than that in the deep ocean model, $\mathcal{C}_5$ and $\mathcal{C}_6$ are set as (1, 0.001). In summary, in this study the standard values of the parameters included in this simple coupled model $(\sigma, k, b, \mathcal{C}_1, \mathcal{C}_2, \mathcal{O}_d, \mathcal{O}_m, \mathcal{S}_m, \mathcal{S}_s, \mathcal{S}_{pd}, \Gamma, \mathcal{C}_3, \mathcal{C}_4, \mathcal{C}_5, \mathcal{C}_6)$

are set as (9.95,28,8/3,0.1,1,1,10,10,1,10,100,0.01,0.01,1,0.001) (e.g., Zhang, 2011a,b; Zhang et al., 2012; Han et al., 2013,2014).

As the study of Han et al. (2014), the fourth-order Runge-Kutta time-differencing scheme is used in this paper to resolve this simple coupled model and the time step equals to 0.01TU (1 TU=100 time steps).

Zhang (2011b) illustrated that, given the model parameters described above, the constructed simple coupled model can effectively simulate a fundamental feature of the real world climate system in which different time scales interact with each other to develop climate signals. Namely, the synoptic to decadal time-scale signals are produced by the interactions between the transient atmosphere attractor, the slow slab ocean and the even-slower deep ocean (see Zhang, 2011a; Han et al., 2014). Again, although the simple coupled model does not have complex physics and cannot consider the issue of impact of

localization and imbalance as in a CGCM, it can help us investigate the fundamental issue we want to address here more directly and clearly.

## 2.2 Ensemble coupled data assimilation

Following Zhang (2011a), during the state estimation, the error statistics evaluated from ensemble model integrations, such as the error covariance between model states, will be used in an ensemble filter to extract observational information to adjust

the model states (e.g. Evensen, 1994; 2007; Anderson, 2001; Hamill et al., 2001; Zhang, 2011a,b; Zhang et al., 2012; Han et al., 2014). In this study, a derivative of Kalman filter (Kalman, 1960; Kalman and Bucy, 1961) called ensemble adjustment Kalman Filter (EAKF, Anderson 2001; 2003; Zhang and Anderson, 2003; Zhang et al., 2007) which is a sequential implementation of ensemble Kalman Filter under an "adjustment" idea is used to implement the CDA scheme. The assumption of independence of observational error allows the EAKF to sequentially assimilate observations into

corresponding model states (Zhang and Anderson, 2003; Zhang et al., 2007). While the sequential implementation provides much computation convenience for data assimilation, the EAKF maintains much of the non-linearity of background flows as possible (Anderson, 2001; 2003; Zhang and Anderson, 2003).

Based on the two-step implementation of EAKF scheme (Anderson, 2001; 2003), the observational increment at an observation location is first computed. The observation is denoted as $\mathcal{Y}$ at time $t$ (simply $\mathcal{Y}$ instead of $\mathcal{Y}_t$) which has the

observation value $\mathcal{Y}^o$ and standard deviation $\sigma_y^o$ (assumed to be Gaussian). Firstly, the reshaping of the model ensemble at the observation location, $\Delta\mathcal{Y}'$ is formulated as:

$$\Delta\mathcal{Y}'_i = \frac{\Delta\mathcal{Y}_i^p}{\sqrt{1 + r_k^2}} \ \ and \ r_k = \frac{\sigma_{k,k}^p}{\sigma_{k,k}^o} \tag{2}$$

where $i$ represents the ensemble index and $k$ denotes the observation index. $\sigma_{k,k}^o$ and $\sigma_{k,k}^p$ are the standard deviation of observation error and its prior estimated ensemble standard deviation, respectively, while $r_k$ is the corresponding ratio. If $r_k > 1$, the ensemble spread is largely reduced by the observation; otherwise, the ensemble remains close to the prior. The

shift of the ensemble mean induced by the observation is computed by:

$$\bar{y}^{u} = \frac{\bar{y}^{\mathrm{p}}}{1 + r_k^2} + \frac{y^{\mathrm{o}}}{1 + r_k^{-2}} \tag{3}$$

We can see that if the prior estimated ensemble standard deviation is greater than that of the observation error, the ensemble mean shifts toward the observation value; otherwise the ensemble mean remains close to the prior model ensemble mean $\bar{y}^{\mathrm{p}}$. Then the observational increment induced by the observation value $y^{\mathrm{o}}$ for the $ith$ ensemble member at the $kth$ observation location is computed as:

$$\Delta y_{k,i}^{o} = (\bar{y}_k^{u} + \Delta y_{k,i}') - y_{k,i}^{p} = \left( \frac{\bar{y}_k^{\mathrm{p}}}{1 + r_k^2} + \frac{y_k^{\mathrm{o}}}{1 + r_k^{-2}} + \frac{\Delta y_{k,i}^{p}}{\sqrt{1 + r_k^2}} \right) - y_{k,i}^{p} \tag{4}$$

Once we get the observational increment at the observation location, then a least square fit is used to distribute the increment over the relevant grid points impacted by the observation using the covariance between the grid index $j$ and the observation $k$, $c_{j,k}^{p}$, using:

$$\Delta z_{i,j} = \frac{c_{j,k}^{p}}{(\sigma_{k,k}^{\mathrm{p}})^2} \Delta y_{k,i}^{o} = \frac{Cov(z_j, y_k)}{(\sigma_{k,k}^{\mathrm{p}})^2} \Delta y_{k,i}^{o} \tag{5}$$

Where $z$ represents a certain state variable at the grid point $j$. The term $\Delta z_{i,j}$ is the contribution of the $kth$ observation to $ith$ ensemble member of the model state estimated at grid point $j$. When an observation is available, the Eq.(5) will be applied to

implement CDA for state estimation in a straight forward manner (Zhang et al., 2007; Zhang, 2011a).

Although many sophisticated inflation algorithms (e.g. Anderson, 2007; 2009; Li et al., 2009; Miyoshi, 2011) exist for atmosphere data assimilation, the inflation scheme for a coupled model is a new subject due to the multiple time-scale nature of the system. Furthermore, trial-and-error experiments show that the usual form of inflation (e.g. only inflate the atmosphere model states or inflate all the model states equally) will lead the analysis to become unstable. Thus, in this paper,

for simplicity and computational convenience as well as convenience for comparison, no inflation is used in our assimilation experiments, just as in Han et al. (2014).

**2.3 Perfect and biased twin experiment setups**

In this study, a perfect twin experiment framework and a biased twin experiment framework are designed, respectively. In both perfect and biased twin experiments, a "truth" model using the standard parameter values listed in section 2.1 is used to

generate the "true" solution of the model states and produce the observations sampling the "truth". Starting from the initial condition (0,1,0,0,0), the "truth" model is firstly integrated forward for 10000TUs (i.e. 1000 model years) for sufficient spin-up and then integrated forward for another 10000TUs to generate the "truth" model states. The observations are produced by sampling the "truth" solution of the model states at an observational interval and superimposing with a white noise simulating the observational errors. As schematically shown in **Fig. 1**, all the observational intervals using in this study are

assumed to be 1 time step (0.01TU). Although in the real climate system, the oceanic observations are usually available less frequently than those in the atmosphere (namely the oceanic observation interval is larger than that we set here), for this

proof-of-concept study we will set the time interval of the oceanic observations as small as possible. The standard deviations of the observational errors are 2 for $\mathcal{X}_1, \mathcal{X}_2, \mathcal{X}_3$ and 0.5 for ω. Also, although the deep ocean lacks observations in the real world, we also conduct some observation simulation experiments for η (the standard deviations of the observational error is 0.06 for η) in this conceptual study.

5    We first want to learn some basics from the perfect experiment which represents an idealized data assimilation regime. In the perfect twin experiment framework, the assimilation model also uses the standard parameter values, but starts from different initial conditions. Using the Gaussian white noises with the same standard deviation as observational errors (2 for $\mathcal{X}_1, \mathcal{X}_2$ and $\mathcal{X}_3$, 0.5 for ω and 0.06 for η) added on the model states at different times during the spin-up run to form the ensemble initial conditions for each ensemble filtering data assimilation experiment. Each assimilation experiment is integrated for 10    10000TUs and only the data obtained in the last 5000TUs is used to conduct error statistics for evaluation. We choose the model states between 9000TUs and 10000TUs during the spin-up at an interval of 50TUs being perturbed to form 20 cases of ensemble initial conditions for each assimilation experiment analysed in section 4 and 5. In this way, we attempt to minimize the dependence of the results of optimal OTWs on ensemble initial states. Then each assimilation experiment will be repeated 20 times starting from these 20 independent ensemble initial conditions and we will analyse the mean value and 15    uncertainty evaluated from these 20 cases.

Then we use the biased experiment setting to simulate the real world scenario. The biased twin experiment framework is similar as the perfect one except that the assimilation model in the biased twin experiment framework has a systematic discrepancy from the observations. Thus, in the biased twin experiment framework, the parameters included in the assimilation model will have 10% errors relative to the standard values. The errors in the parameters will be the only model 20    error source.

**Fig. 1** also illustrates the assimilation update intervals (the assimilation intervals are 5 time steps for atmosphere, 20 time steps for slab ocean in all assimilation experiments and 100 time steps for deep ocean if using the η observations) as well as the length of observational time window (OTW), which will be used throughout the study. In addition, the coupling strength between the atmosphere and ocean may have influences on the characteristic variability time scale of each coupled medium, 25    so as on the optimal OTW. We discuss this issue through changing the values of coupling coefficients $\mathcal{C}_1$ and $\mathcal{C}_2$. In this simple model case, the model stability is sensitive to coupling coefficient $\mathcal{C}_1$(Zhang et al., 2012) and changing $\mathcal{C}_1$ only influences on the chaotic component, so here we just change $\mathcal{C}_2$ to investigate the impact of coupling coefficient between the atmosphere and upper ocean on the optimal OTWs. As in Zhang and Anderson (2003), an ensemble size of 20 is applied in all assimilation experiments in this study.

## 3 The length of observational time window (OTW) and retrieval of characteristic variability

### 3.1 Influence of OTW on the accuracy of CDA

In order to exhibit the influence of OTW on the quality of climate analysis, we show 3 simple assimilation experiments (the time series of $\omega'$s absolute errors) in **Fig. 2**: 1) standard CDA (green) (assimilating the observations rights at the analysis times without any atmospheric/oceanic OTW); 2) OCN-OTW(5) CDA (red) (assimilating all 11 ocean observations in an oceanic OTW with a half-width of 5, defined as the length of the OTW hereafter, but no atmospheric OTW is considered) and 3) OCN-OTW(100) CDA (blue) (assimilating all 201 ocean observations in an oceanic OTW with the length of 100, but no atmospheric OTW is considered). Three assimilation experiments above are all conducted using the perfect model setting (all the parameters use their standard values) and the uni-variate adjustment scheme. The atmospheric and oceanic update intervals are 0.05 TU and 0.2 TU, respectively. While the standard CDA does not use the atmospheric and oceanic OTWs and only assimilates the observations right at the analysis time, the OCN-OTW CDA incorporates all the valid observations collected in the oceanic OTW. All the three assimilation experiments above do not use an atmospheric OTW.

From **Fig. 2**, we can see that a small OCN-OTW (total 11 observations in the oceanic OTW) can make a much better ocean analysis than the standard CDA (comparing the red line with the green line). We can understand that this is because an OTW can provide more observational information thus enhancing the observational constraint so as to improve the accuracy of climate analysis. However, comparing the blue line to the green/red line, it is clear that a too large OTW degrades the quality of the ocean analysis. The results of these simple assimilation experiments tell us, if an appropriate OTW is used, we can gain optimal climate analysis. How can we determine such an optimal OTW? Next, starting from analysing characteristic variability of each coupled medium, we will discuss the methodology how to determine an optimal OTW for each medium in a coupled climate system.

### 3.2 The time scale of characteristic variability and optimal OTW

The key to improve the accuracy of climate analysis in CDA is accurately recovering characteristic variability of different media in the coupled system. Thus we can assume that the length of optimal OTW for each medium will have some relationship with the corresponding characteristic variability time scale. Then, we should first analyse the time scale of characteristic variability in each medium.

**Fig. 3** presents the power spectrum of $\mathcal{X}_2$, $\omega$ and $\eta$ based on the model states with a 4800-TU length (totally 480000 data) after the spin-up described in section 2.3. From **Fig. 3**, we learned that in this simple model, the characteristic variability time scales of atmosphere ($\mathcal{X}_2$), upper ocean ($\omega$) and deep ocean ($\eta$) are about 1-2 TUs (1-2 model months), 50-100 TUs (5-10 model years) and 500 TUs (5 model decades), respectively. Namely, the characteristic variability time scale of the slab ocean is much larger than that of the atmosphere but smaller than that of the deep ocean.

An optimal OTW aims to provide maximal observational information that best samples characteristic variability of that medium during the data blending process. Thus the length of the optimal OTW should be smaller than the corresponding

characteristic variability time scale, which means that the optimal OTW in the atmosphere must be much smaller than 1 TU (100 time steps), and in the ocean, the optimal OTW must be much smaller than 50 TUs (5000 time steps). If we will take observations for η, the optimal OTW for η must be much smaller than 500 TUs (50000 time steps). From **Fig. 3**, we also see that the characteristic variability time scales of different coupled media are a little larger than the corresponding ones set in the Eq.(1). This is owing to the strong nonlinearity and smoothness of the fourth-order Runge-Kutta time-differencing scheme that prolongs the characteristic variability time scales of the simple coupled model. But they do not change the essence of the problem we address in this study. Given different time scales of characteristic variability in different media, in the following section we will further detect the optimal OTWs based on the corresponding characteristic variability time scales and examine their impact on the quality of climate analysis in CDA.

## 4    Detection of the optimal observational time window

In this section, with the perfect model framework described in section 2.3, we first conduct a series of CDA experiments with different ATM-OTWs and different OCN-OTWs to detect the optimal OTW for each medium. The assimilation scheme is the simple uni-variate adjustment scheme serving as a proof-of-concept study. To eliminate the dependency of results on initial states, each experiment is repeated 20 times starting from the 20 independent initial conditions described in section 2.2. Then the mean value and the spread of 20 cases' RMSEs are plotted in **Fig. 4**.

**Fig. 4a** shows that the optimal ATM-OTW is 1, i.e., the optimal ATM-OTW includes only 3 atmosphere observations, with which the assimilation produces the lowest RMSE of the atmosphere and the smallest spread (In this study each assimilation experiment will be repeated 20 times starting from 20 different independent initial ensemble conditions. Here the spread just represents the standard deviation of these 20 cases' results. Thus it will be smallest when using the optimal OTW.). In these experiments for detecting the optimal ATM-OTW, the ocean assimilation is kept as the standard setting (i.e. no OTW, 0.2 TU update interval). Then we keep the ATM-OTW as 1 and change the length of OCN-OTW to produce **Fig. 4b**.

From **Fig. 4b**, we can see that the optimal OCN-OTW is about 10 (i.e., each OTW includes total 21 observations), with which the lowest ω-RMSE and the smallest RMSE spread are produced. Compared to the case of standard CDA (denoted as CDA_NOTW), the uses of optimal ATM-OTW and OCN-OTW make the RMSEs of $\mathcal{X}_{1,2,3}$ and ω significantly reduced. When the RMSE of ω has a distinguishable sensitive variation with respect to OCN-OTWs, the RMSE of $\mathcal{X}$ (Because we just choose the optimal OCN-OTW from the figure of ω-RMSE. Thus in this study the variation of $\mathcal{X}$-RMSEs in the OCN-OTW space is not shown.) does not show such a sensitivity to the optimal OCN-OTW. This means that in this simple system, due to the strong nonlinearity and chaotic nature of the "atmosphere", the improved accuracy for ω from optimal observational constraint is not sufficient to impact the "atmosphere" (this point will be expanded in section 5.3). Similar to the characteristic variability time scale of the slab ocean vs. that of the "atmosphere" shown by **Fig. 3**, the optimal OCN-OTW is much larger than that of ATM-OTW.

To further understand the relationship between the optimal OTW and characteristic variability time scale, we also examine the η-RMSEs in the space of η-OTWs. The assimilation interval of the pycnocline depth is set as 1 TU (100 time steps), which is much larger than that of the slab ocean. When we change the η-OTW, the optimal ATM-OTW and OCN-OTW detected from **Figs. 4ab** are used. As shown in **Fig. 4c**, the optimal η-OTW is about 100 (i.e. total 201 observations), which is much larger than that of OCN-OTW and smaller than the characteristic variability time scale of the deep ocean pycnocline depth. With the optimal η-OTW, the RMSE of η is reduced about 77.4% from the level of the CDA_NOTW.

We also check the variation of the 20-case mean ensemble spread in the space of OTWs as shown in **Fig. 5.** The mean and standard deviation of the ensemble spreads of $\mathcal{X}_{1,2,3}$ and $\omega$ (the uncertainty of the state estimation in each assimilation experiment is shown as the blue shadow in **Fig. 5**) gradually decrease when the ATM-OTW and OCN-OTW become larger. When the OTWs are set too large (here the ATM-OTW and OCN-OTW are greater than 20 and 250, respectively), the ensemble spreads of $\mathcal{X}_{1,2,3}$ and $\omega$ decrease dramatically. This is owing to that when we increase the length of the OTW, more observations will be included in the OTW and then assimilated into the corresponding model states, which can function as a smoother. The longer the lengths of OTWs are, the stronger the smoother will be. Also under this circumstance, the too strong smoother will distort characteristic variability of the model states, which explains the blue line of **Fig. 2**. From **Fig. 4** and **Fig. 5,** we can see that the mean of ensemble spread is significantly smaller than that of the corresponding RMSE. It is owing to that no inflation scheme is applied in this study. And the statistics for evaluation are conducted from the data obtained in last 5000TUs. Thus after first 5000TUs' assimilation in each assimilation experiment, the ensemble spreads of model states have been greatly reduced due to no inflation. Then the mean ensemble spread is significantly smaller than the mean RMSE.

To understand the essence of optimal OTWs, we show the auto-correlation for each model state and marks the time correlation coefficients at the time scales of optimal OTWs for $\mathcal{X}_2$ (panel *a*), $\omega$ (panel *b*) and η (panel *c*) detected from **Fig. 4** in **Fig. 6**. The result is the mean of 20 cases. In each case, the number of data is 10000 steps (100TUs), which are chosen from the period of 5000TUs to 9000TUs in the truth run after spin-up. From **Fig. 6** we can see that all auto-correlation at the optimal OTW length are located around 0.995. This means that the observations included in an optimal OTW are extremely highly correlated with the model state at the analysis time. This can be understood since in this sequential assimilation scheme all the observations included in an OTW are assumed to be sampled at the analysis time so that the difference among them must be in a negligible range. Under such a circumstance, the optimal OTWs provide maximal observational information that best fits characteristic variability and minimizes the analysis error.

## 5    Influences of realistic assimilation scenarios on optimal OTW

In this section, we first show the impact of the multi-variate adjustment scheme on the optimal OTWs in perfect model setting. Then we discuss the influence of model bias through a biased model framework. We will also investigate the impact of coupling strength on the optimal OTWs.

## 5.1 Influence of multi-variate adjustment on optimal OTWs

While the experiments with the uni-variate adjustment scheme provide us a direct understanding of the influence of the OTWs on CDA, we want to check whether or not it also applies to the multi-variate adjustment scheme. So we repeat the experiments described in section 4 but with the multi-variate adjustment scheme. The results are shown in **Fig. 7**. Here the multi-variate adjustment scheme is only limited to the atmospheric observations (i.e., only the cross-covariances among $\mathcal{X}_1, \mathcal{X}_2, \mathcal{X}_3$ are used) (as indicated in Han et al., 2013, the multi-variate adjustment scheme using the coupling cross-covariance between different coupled media involves complex scale interactions and may complicate the investigation of the problem we are addressing here). The results shown in **Fig. 7** are similar with that in **Fig. 4**, suggesting the multi-variate adjustment scheme has little influence on the optimal OTWs, since it does not change the characteristic variability time scales (especially in this simple model).

The perfect experiment framework provides a direct guideline for the relationship between the optimal OTW and the corresponding characteristic variability time scale. However, in reality, the numerical model has errors and is biased with the observation. It is necessary to investigate the influence of model bias on optimal OTWs so as on the quality of CDA.

## 5.2 Influence of model bias on optimal OTWs

With the biased model experiment framework described in section 2.3, we repeat all the experiments above for detection of the optimal OTWs. The results are shown in **Fig. 8**. Compared to the results in the perfect model setting, the results in the biased model setting have 2 differences. First, the optimal ATM-OTW and OCN-OTW are larger than their counterparts in the perfect model setting, becoming 3 and 20 (namely the total observations are 7 and 41, respectively). Second, the RMSE curves in the space of OTWs show more concavity and sensitive variation. This is more distinguishable in the curve of ω-RMSEs in the OCN-OTW space. All these phenomena can be explained by the influence of model bias on the assimilation quality. On the one hand, due to the existence of model bias, the assimilation not only needs observations to fit the observed variability but also needs observations to reduce the mean discrepancy between the model and observation. This requires stronger observational constraints. An optimal OTW that makes the smallest RMSE of model states must include more observed data. On the other hand, the forecast ensemble in a biased model underestimates the forecast error, which results in that the EAKF under-weights the observations. Therefore the optimal OTWs are larger than those in the perfect experiment case that the observations included in the optimal OTWs will be assimilated for multiple times, which results in improving of filter performance. The test experiment for the optimal η-OTW is also consistent with this point (in **Fig. 8c**): the optimal η-OTW in the biased model setting is larger than that in the perfect model setting. Then we also investigate the influence of OTWs on the quality of CDA with the multi-variate adjustment scheme in the biased experiment framework (not shown here). Results are the same as the perfect model setting case, i.e., multi-variate adjustment scheme does not change the optimal OTWs.

Comparing the results from two experiment frameworks, we can see that regardless of perfect or biased model setting used in the assimilation experiments, the optimal OTW must be associated with the corresponding characteristic variability time scale in the medium. It is clear that while using observations in an OTW increases observational information, a too large OTW can distort the characteristic variability of coupled media during the information blending process. Therefore choosing an optimal OTW that is much smaller than the medium's characteristic variability time scale is very important. The simple model results suggest that the length of an optimal OTW is about 1-5% of the medium characteristic time scale, with which characteristic variability of the medium can be retrieved most accurately.

In this study, the OTW validates the observations in a time window to the analysis time and all the observations included in the OTW are sequentially assimilated with their original error scales. Another general approach is to assimilate the average of the observations included in the OTW but the observational errors decrease as $1/\sqrt{N}$ of their original error scales ($N$ represents the number of the observations included in the OTW). From the comparison of these two methods (not shown), we can see that the results obtained by them are almost the same. From the perspective of the calculation process of the EAKF method, owing to that no inflation scheme was used, after many assimilation steps the ensemble spreads of the models states have been greatly reduced and significantly smaller than the corresponding observational error scales. And the prior ensemble member will be very close to the prior ensemble mean. Thus the analysis adjustments obtained by these two methods will be almost the same. It is worth mentioning that although the resulting RMSEs obtained by these two assimilation schemes will be different when using the suitable inflation schemes, the lengths of the optimal OTWs are still the same and the essence of this study still firms and does not change.

Also among above assimilation experiments in this study, we have not considered the temporal offset induced by the difference between the time of observations in the OTW and the analysis time. Here we can use the de-correlation coefficients to weight the observations included in the OTW and avoid overweighting them. The comparison of these two assimilation approaches (non-weighted and weighted) have been conducted (the results are not shown). From the comparison we learn that the lengths of the optimal OTWs obtained by these two assimilation schemes are similar except that the RMSEs in the weighted observation experiment will be lower than that in the non-weighted one when using longer OTWs (when the length of ATM-OTW is greater than 4 and/or that of the OCN-OTW is larger than 50). This is owing to the high correlation between the observation included in the optimal OTWs and model states at the analysis time (exceed 0.995). Thus the influence of the temporal offset can be ignored and the results obtained by these two schemes shall be almost the same when using the shorter OTWs. When we use the longer ones, the correlation will decrease and influence of the temporal offsets will be obvious that the results of the weighted observation experiment will be better. For the CDA systems in the CGCMs, owing to the complex physics and dynamics, the influence of the time offsets will be obvious and the weights of the observations will be very necessary. But from this simple model case, we can see that whether or not using the weighted observations, the relationship between the characteristic variability time scales and the optimal OTWs will be robust and the essence of this study is established.

## 5.3 Influence of coupling strength on optimal OTWs

Changing the coupling strength (controlled by the coupling coefficients $\mathcal{C}_1$ and $\mathcal{C}_2$ in this case) between the atmosphere and upper ocean may have some influence on the characteristic variability time scales of coupled media, so as on the optimal OTWs. Test experiments show that changing the coupling coefficient $\mathcal{C}_1$ has little influence on the characteristic variability time scales of $\mathcal{X}_{1,2,3}$ and $\omega$. This is because the characteristic time scale of $\mathcal{X}$ is determined by the chaotic nature of the Lorenz equations, not by the oceanic forcing associated with the coupling coefficient $\mathcal{C}_1$. Therefore, here we just change $\mathcal{C}_2$ to investigate the coupling coefficient between the atmosphere and upper ocean on the optimal OTW of $\omega$. Setting the values of $\mathcal{C}_2$ as 1.5, 1.25, 1.0, 0.8, 0.5 and 0.1 and keeping $\mathcal{C}_1$ as 0.1, we repeat all the biased CDA experiments with the multi-variate adjustment scheme. The results are shown in **Fig. 9**, which presents the power spectrum of $\mathcal{X}_2$ and $\omega$ (panel $a$ and $b$) of six cases above based on the model states between 5000 TUs and 9800 TUs, as well as the time series of model states between 5000 TUs and 5100 TUs (panel $c$) after the spin-up described in section 2.3. We can see that changing $\mathcal{C}_2$ does not influence on the characteristic variability time scale of the atmosphere but strongly influences on variability of the slab ocean. From the equation of $\omega$, the characteristic variability time scale of $\omega$ is determined by the combination of the atmospheric forcing and the periodic external forcing. When $\mathcal{C}_2$ is small, the forcing of atmosphere to ocean is weak and then the periodic external forcing plays a dominant role on determining the characteristic variability time scale of ocean component.

Then we examine the difference of the optimal OTW of $\omega$ in the six cases above, as shown in **Fig. 10**. The results show that changing $\mathcal{C}_2$ does not have any influence on the optimal ATM-OTW (not shown). From Panels $a$ and $b$ we can see that when $\mathcal{C}_2$ is smaller, the optimal OCN-OTW is larger. This can be explained by the increasing role of the periodic external forcing on determining variability of the slab ocean, for which data assimilation needs more observational information to recover the periodic variation of $\omega$, determined by the time scale defined by $\mathcal{S}_{pd}$(10 TU). When $\mathcal{C}_2$ is larger than 1.0, changing it has little influence on characteristic variability of the $\omega$, so as on the optimal OCN-OTW.

On the one hand these experiments can further illustrate the idea that a close relationship between the length of the optimal OTW and the corresponding characteristic variability time scale exists. On the other hand, for a realistic CDA system, the coupling physics could be very complicate and affected by many factors. The results of this simple model give the insights that when determining the length of the optimal OTWs for a realistic CDA system, we can only consider such factors that have obvious influence on the characteristic variability time scales. In this way, the process of determining the optimal OTWs in a realistic CDA system can be greatly simplified and make it possible to apply the method of using the optimal OTWs into the realistic CDA system.

## 6    Summary and discussions

With a simple conceptual climate model and the EAKF method, the impact of OTWs on the quality of CDA has been investigated in this study. This simple conceptual coupled model consists of a synoptic atmosphere (Lorenz, 1963) and

seasonal-interannual slab upper ocean (Zhang et al., 2012) coupling with a decadal deep ocean (Zhang, 2011a,b), and reasonably simulates the typical interactions between multiple time-scale components in the climate system. Determined from the characteristic variability time scale in each coupled medium, an optimal OTW provides maximal observational information to best fit characteristic variability of the medium during the data blending process. With correct scale interactions within the coupled system, CDA can recover the climate signals most accurately through incorporating all observations in the optimal OTWs into the coupled model. Although in an idealized and simple model circumstance, the conclusion addressing best fitting characteristic variability in each medium with the optimal OTW is comprehensive and therefore provides a guideline for improving climate analysis and prediction initialization when real observations are assimilated into a CGCM. For example, learned from the simple model results, we may consider to improve the quality of climate analysis and prediction initialization through accurately recovering some important characteristic variability in the atmosphere (sub-diurnal variations, for instance) and ocean (diurnal cycle in the tropical oceans, for instance).

However, the current work only can serve as a proof-of-concept study. Although CDA with the optimal OTWs has shown promising improvement in this simple model, serious challenges still exist for detecting optimal OTWs in the real world with a CGCM for improving climate analysis and prediction. First, the characteristic variability time scales in different media of the real world are complex and it remains great challenges to identify the characteristic variability of the different component models and the real atmosphere, upper and deep ocean, which need to be further studied. Also in a real ocean model, the upper and deep ocean is inseparable, which bring some troubles to use different OTWs for different parts of the same ocean model. Second, due to model biases, characteristic variability in a CGCM may be different from the real world. The combination of variability of the real world and that of the model may further complicate the problem. Therefore, model bias and its influence on model variability need to be thoroughly analysed before an optimal OTW is determined. Thirdly, the coupling physics between different coupled components are very complicate and impacted by many factors for a realistic CDA system. Even though we only consider the factors which will obviously impact the characteristic variability time scales when determining the length of OTWs for different coupled components, it remains as heavy workload. In addition, in this study we assume that all observations in the OTWs have equal weights to contribute to the observational constraint. In the real observation case, the observation far away from the assimilation time should have less contribution to the state estimation at the assimilation time. How to take the time correlation into account in a sequential algorithm needs to be studied before implementing optimal OTWs into the assimilation with CGCM and real observations.

**Acknowledgements**

This work was supported by the National CMOST Key research & development projects 2017YFC1404100, 2017YFC1404102, the NSFC (No.51379049, 41676088, 41775100), the Fundamental Research Funds for the Central Universities of China (No.HEUCFX41302, HEUCFD1505, HEUCF160410), the Young College Academic Backbone of Heilongjiang Province (No.1254G018), the Scientific Research Foundation for the Returned Overseas Chinese Scholars,

Heilongjiang Province (No.LC2013C21), and Harbin Engineering University and China Scholar Council (awarded to Xiong Deng for two and half years' study abroad at UW-Madison – NOAA/GFDL Joint Visiting Program). We thank Drs. Liwei Jia, Wei Zhang, Xuefeng Zhang, Wei Li, Lianxin Zhang and Shuo Yang for their comments and suggestions at the early version of this manuscript. Also special thanks to three anonymous reviewers for their critical comments that contributed to great

improvements in the original manuscript.

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

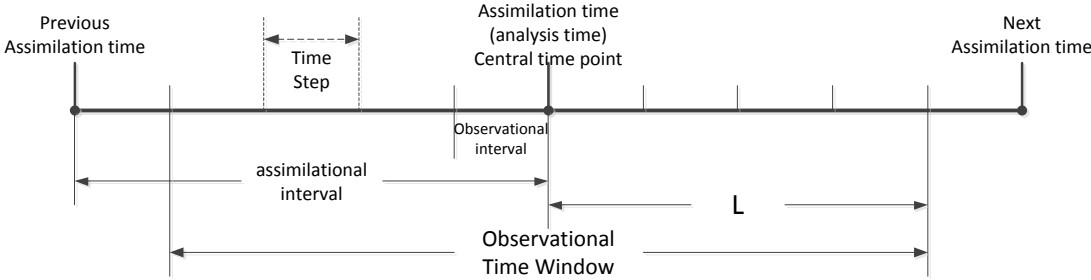

**Figure 1:** The schematic for the assimilation interval, the length of observational time window (OTW) as well as observational interval in terms of the model integration time step. Here L represents the time steps at one side of OTW. For example, OCN-OTW (L) in the content stands for an ocean observational time window with total observations of 2L+1.

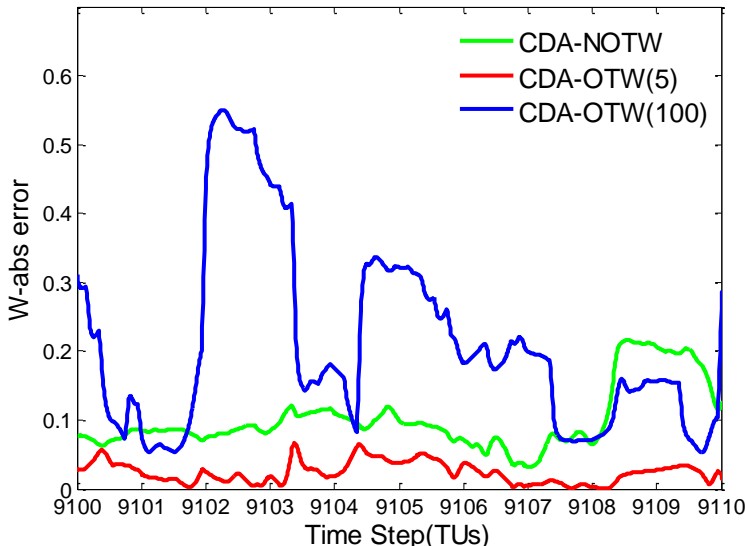

**Figure 2:** Time series of the absolute errors of the slab ocean variable ($\omega$) in 3 assimilation experiments based on the model states between 9100 TUs and 9110 TUs assimilation results in the perfect model experiment framework with the uni-variate adjustment scheme. Green – CDA control with the standard update intervals of 0.05 TU for $\mathcal{X}_{1,2,3}$ and 0.2 TU for $\omega$; Red – CDA with an ocean observational time window (OCN-OTW) of 5 time steps [OCN-OTW (5)]; Blue – CDA with OCN-OTW (100).

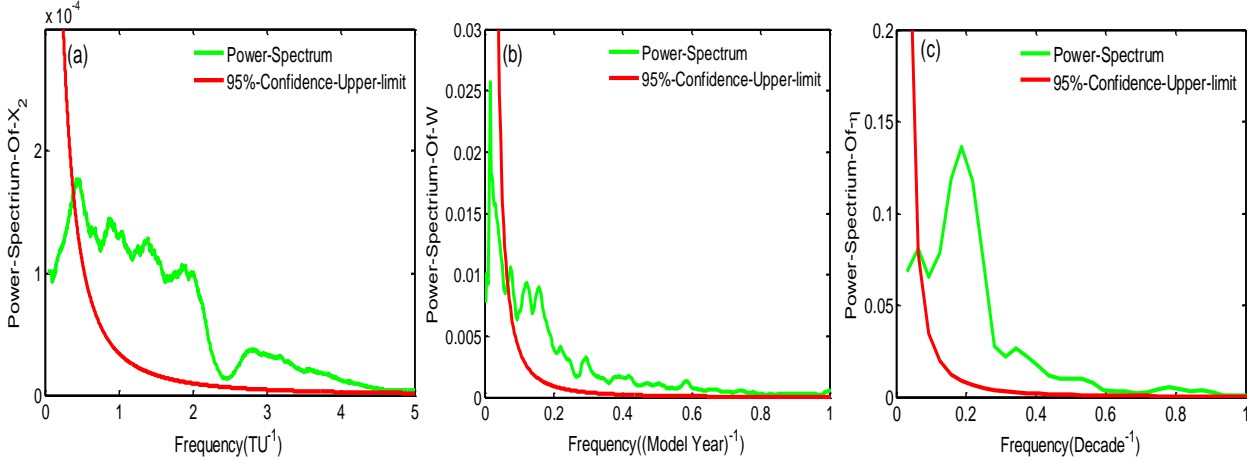

**Figure 3:** The power spectrum (green) of (a) $\mathcal{X}_2$ (b) $\omega$, (c) $\eta$ based on the model states between 5000TUs and 9800TUs integrations after the spin-up which integrates for 10000TUs from the initial condition (0,1,0,0,0) with respect to the frequency, with 95% statistics significance (red).

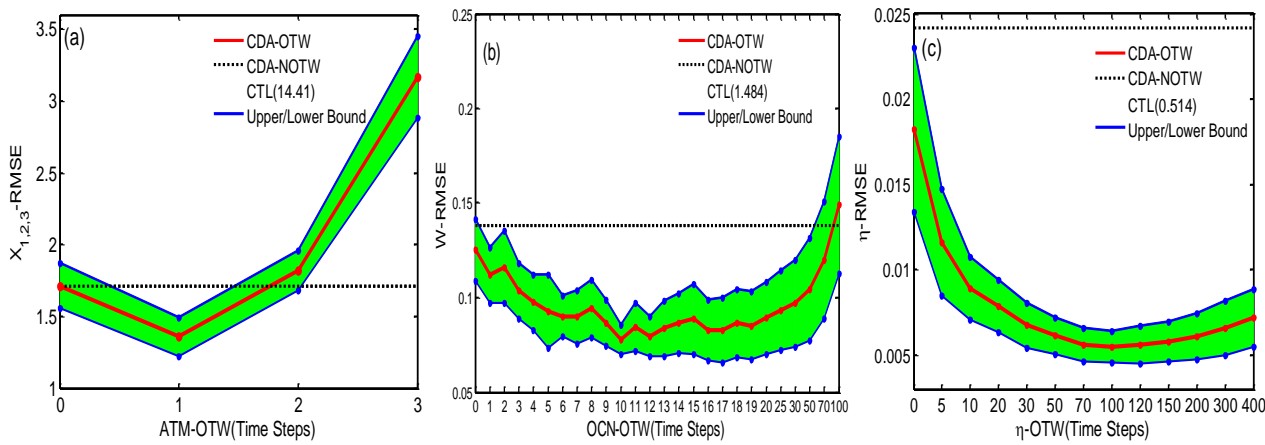

**Figure 4:** Variations of root mean square errors (RMSEs) of (a)"atmospheric" states $\mathcal{X}_{1,2,3}$ (namely the average of $\mathcal{X}_1$, $\mathcal{X}_2$ and $\mathcal{X}_3$ RMSEs) in the space of ATM-OTW length when the "oceanic" state ($\omega$) only uses a single observation at the assimilation time; (b) "upper ocean" state ($\omega$) in the space of OCN-OTW length when the ATM-OTW is fixed 1 as shown in panel (a) (1 for the ATM-OTW, i.e. 3 observations in each window, see the caption of Fig. 1) but the OCN-OTW (for $\omega$) is varying and (c) "deep ocean" state ($\eta$) in the space of $\eta$-OTW length when the "deep ocean" observations are assumed to be valid and the ATM-OTW and OCN-OTW are fixed as 1 and 10, respectively. The experiments are conducted in a perfect model setting with a simple uni-variate adjustment scheme. The red lines are the 20-case mean, each using different initial conditions taken from different periods in the control integration (see description in section 2.2), and the blue lines represent the upper/lower bounds (mean $\pm$ standard deviations) of the RMSEs. An OTW with the length of 0 represents only assimilating the observation at the assimilation time (i.e. with no OTW, dashed-black lines). The RMSE values of the control case (no observational constraint, called CTL) are marked in the parenthesis.

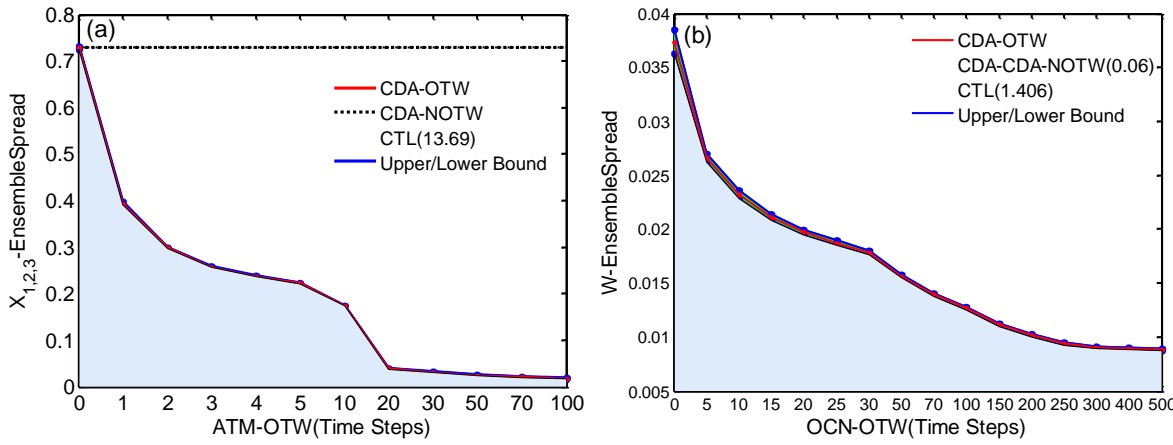

**Figure 5:** Same as the panels (a) and (b) in Fig. 4 but for the variation of ensemble spreads of the model states. In panel (b) the optimal ATM-OTW is also set as 1. The area between the lower and upper bounds (blue) represents the range evaluated from the 20 cases. And the blue shadow below the ensemble spreads represents the range of the uncertainty of state estimation in each assimilation experiment.

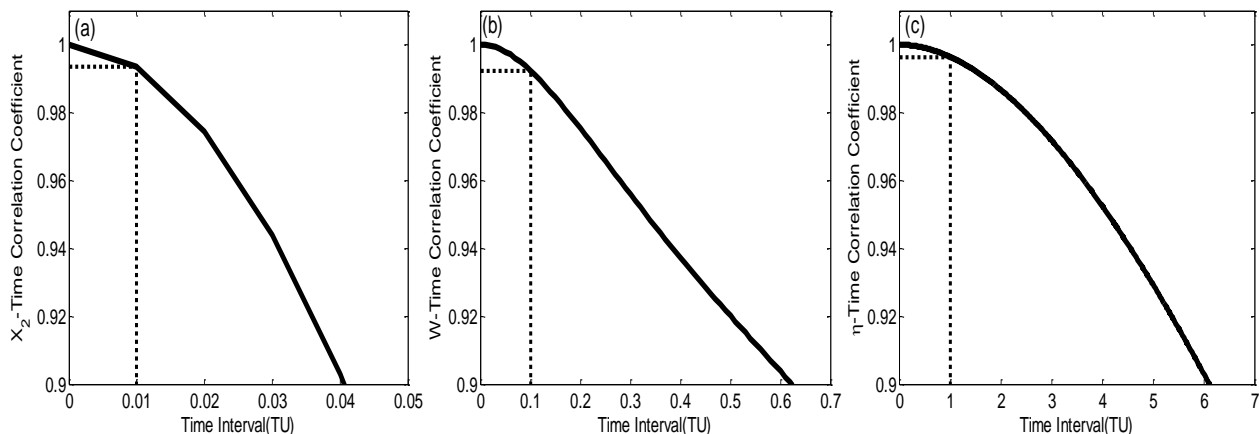

**Figure 6:** The auto-correlation coefficient of (a) $\mathcal{X}_2$ (b) $\omega$, (c) $\eta$ in the space of lag times are marked by corresponding time correlation coefficients at the time scale (L) of optimal OTWs as detected by Fig. 4 for different media (The black dashed lines). What are shown is the mean of 20 cases. In each case, an independent section (each has 10000 data of the state – 100 TUs with the interval of 0.01 TU) is
10   used to evaluate the lag correlation coefficient. The 20 independent sections are taken from the model states apart each 200TUs between 5000TUs and 9000TUs integrations after the spin-up of 10000TUs from the initial condition (0, 1, 0, 0, 0).

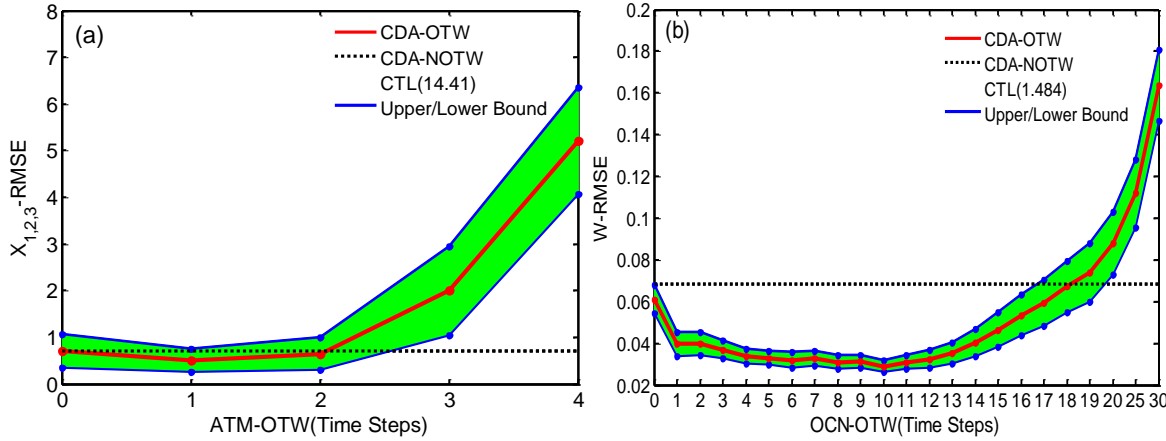

**Figure 7:** Same as Fig. 4 but using multi-variate adjustment scheme. In panels (b) the optimal ATM-OTW is also set as 1.

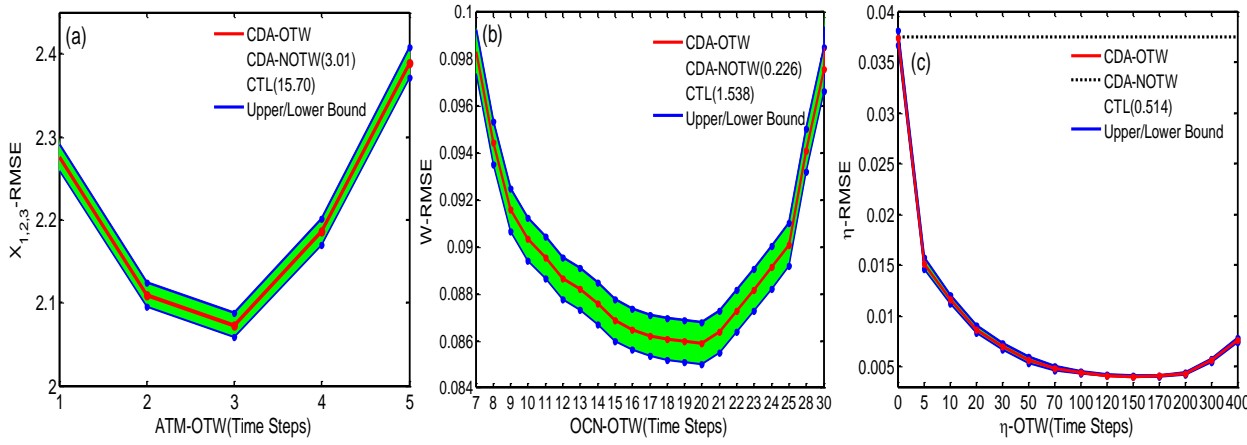

**Figure 8:** Same as Fig. 4 but using the biased model setting. In panel (b) the optimal ATM-OTW is set as 3. And in panel (c) the optimal ATM-OTW and OCN-OTW are kept as 3 and 20, when the "deep ocean" observations are assumed to be valid.

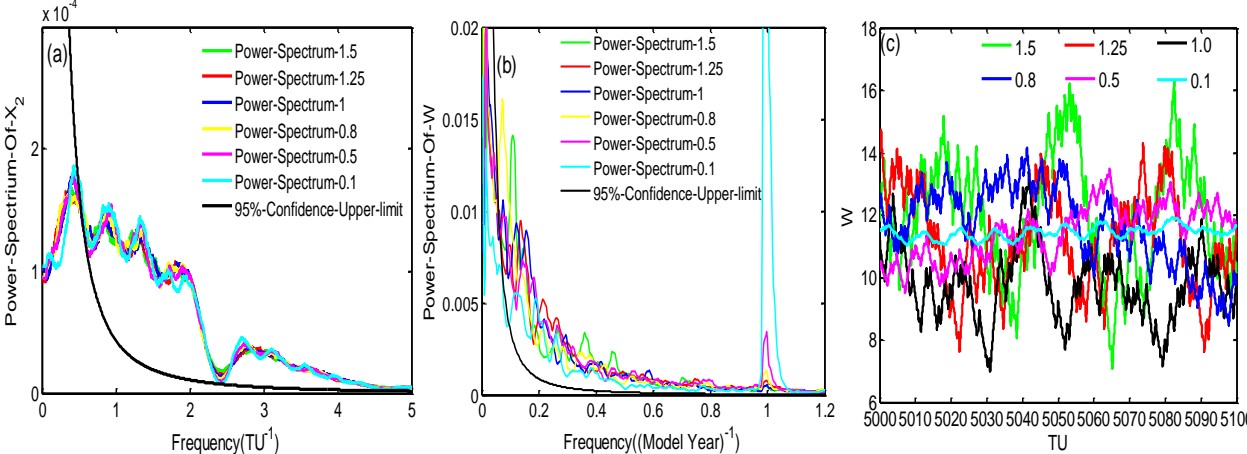

**Figure 9:** The power spectrum of (a) $\mathcal{X}_2$ and (b) $\omega$ based on the model states between 5000 TUs and 9800 TUs integrations after the spin-up which integrates for 10000TUs from the initial condition (0,1,0,0,0) with different coupling strength ($\mathcal{C}_2$ is set as 1.5, 1.25, 1.0, 0.8, 0.5 and 0.1 while $\mathcal{C}_1$ keeps as 0.1). Panel (c) shows the time series of the model state $\omega$ between 5000TUs and 5100TUs integrations corresponding to the six cases.

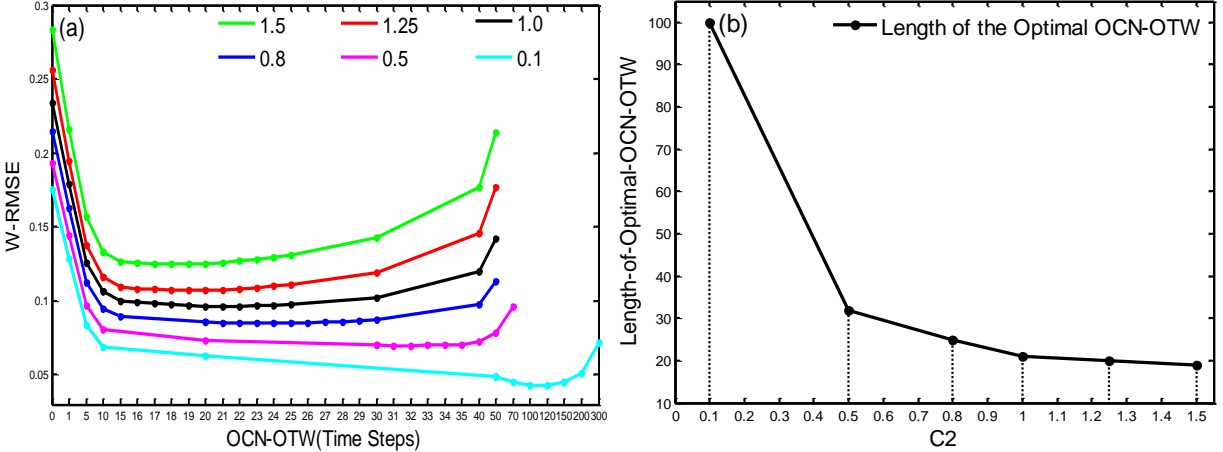

**Figure 10:** Panel (a) is same as panel (b) in Fig. 9 but for using six different coupling strength cases (with $\mathcal{C}_2$ values as 1.5, 1.25, 1.0, 0.8, 0.5, 0.1 while $\mathcal{C}_1$ keeps as 0.1). Panel (b) is the variation of the length of the optimal OCN-OTW with respect to the values of $\mathcal{C}_2$.