# Peer review of "Impact of Observational Time Window on Coupled Data Assimilation: Simulation with a Simple Climate Model"

_Nonlinear Processes in Geophysics, 2016_

## Referee Comment (RC1) · Anonymous Referee #1 · 17 Feb 2017

This study investigated the impact of observational time window (OTW) on coupler data assimilation using a simple climate model. The topic is interesting. It has shown a lot of interesting results. The experiments were well designed. The structure of the paper is good too. I can receive the big picture of the study. However It need some improvement on the details and presentation.

1 the title do not fit. You are not discuss the "Impact of Optimal Observational Time Window' but "Impact of Observational Time Window". It may be " Impact of Observational Time Window in Coupled Data Assimilation with a Simple Climate Model"

2 The concept of OTW in this study validates the observations in a time window to the center of the window ( analysis time). It is very useful technique for data assimilation

(DA). More or less it has been applied in the assimilation with real observations. Here the OTW in this study is applied in 3-dimension DA but not 4-dimension DA, which need address in the introduction. The citations of OTW are not very relevant (Page 2, line 13-15). I could not find the clear concept of OTW and how they applied in data assimilation. SO you should address the technique details in the paper. I guess that you treat all the observations independently and assimilate then sequentially using their original error scales. another approach is to assimilate the average values of the observations in the OTW. It is worth to compare these two approach in the study.

3 From your result I can not connect the de-correlation time with the optimal OTW directly. The optimal OTW is much short than the de-correlation time. Like in figure7, the correlation is 0.995 for the optimal OTW. Please revise the abstract and conclusion. The time scales of the variables really can not provide the useful information to quantify the optimal OTW. I suggest you remove or shorten the discussion and analysis related to the time scales of the variables.

4 The time (s) of observations in the OTW are different from the assimilation time. The temporal offsets introduce the represent errors for the observations, which need to be consider during analysis. If not, EnKF overweights the observations The analysis will not be optimal and the analysis ensemble spread will has negative biased. Please address this part and justify your assimilation.

4 It is hard to find the assimilation intervals for different components. Please address them together in experiment setup (2.3). I noticed that the OTW is longer than the assimilation interval, which means that you assimilate the same observations multiple times. In the perfect framework, it should degrade the filter performance from my understand and the information theory. Can you explain why you can achieve a good analysis under that circumstance. For a biased model framework, people do use observation multiple times to compensate the negative bias of forecast ensemble spread. So it make sense to see the optimal OTW in biased model cases are longer than that in perfect model case.

5 The section 2.2 of "Ensemble coupled data assimilation" is incomplete. People will not understand your equations without trace back to your references. Please provide the completed two steps of EAKF.

6 Page 9 the last paragraph. The discussion of OTW with 4D-var is very confusion and hand-waving. The OTW in 4D-var is very different concept than yours. You should remove it.

7 Figure 4(a & b) show that the lower bound of RMSE reaches to 0. That does not make sense. Please check.

8 Figure 4 and Figure 5 show that the ensemble spread is significantly smaller than the RMSE. Please address the reasons and the effect on the assimilation.

9 P11 Lin13 I can not buy the statement of " since more observational information is needed to compensate the model bias and recover the characteristic variability." The reason should be " the forecast ensemble in a biased model underestimates the forecast error, which results the EnKF underweight the observations. Therefore one can improve filter performance by using observation multiple times"

10 A reading proof is required to improve the manuscript. There are too many typos and grammar errors. For example Page 2 Lin 33 The "ensemble filter" should be "ensemble Kalman filter" Page 6 Lin 13 "but they are started from different initial states." can be deleted Page 6 Lin 15 "observation" should be "truth" Page 6 Lin 33 "the coupling co-efficient $C_1$ is sensitive to model stability" should be " the the model stability is sensitive the coupling coefficient $C_1$" Page 7 Lin3-5 The whole sentence need rephrase. You can just mention that " Following Zhang and Anderson (2003) an ensemble size of 20 is applied in all experiments in this study." Page 7 Lin9-11 "absolute error" is not under-standable. I guess that is " the absolution values of error" . The experiments are not described clearly. Please rewrite. Page 7 Lin28 the sentence is not completed. Page 7 Lin 30 Please also give the equivalent time of atmosphere (1-2 model month?). The deep ocean should be 50 model year.

---

## Referee Comment (RC2) · O. Martinez-Alvarado (Referee) · 22 Feb 2017

General comments

The article presents results relevant to coupled data assimilation for systems, such as the ocean-atmosphere system, in which there is a large dissimilarity between the timescales that characterise each one of the subsystems (called media in the paper referring to the atmosphere and ocean). In particular, the study focuses on the impact of changing the length of the observational time window (OTW) for each of medium and present arguments linking the optimal OTW to the characteristic variability time scale in each coupled medium. The results are significant and the presentation is mostly clear and concise. However, the paper would benefit from proof-reading focusing on English

language aspects that would make it even clearer. I have included some comments referring to this point, but I did not try to compile an exhaustive list.

Specific comments

1 It would be useful to have more background information on the CDA strategy studied in the paper. In particular I was wondering why the observational data should be assumed at the assimilation time when 4D Var would avoid that problem. It was only in Section 4 that 4D Var was mentioned as an alternative.

2 P2, L20-22: The two questions posed are circular in the sense that the answer to the first would depend on the effects of varying the OTW on the quality of CDA and the answer to the second depends on the existence of an optimal OTW. I would suggest the following rearrangement: '1) What is the impact of varying OTWs for each coupled component within the coupled model framework on the quality of CDF for climate estimation and prediction initialisation? 2) Based on this impact, Is there an optimal OTW so that assimilation fitting has maximum observational information, but minimum variability distortion?' The point is subtle, but it might be worth doing for clarity.

3 Equation (1): Should there be a dot over eta? Is the dot over C6 correct?

4 It is not necessary to describe the Runge-Kutta method in Section 2.1.

5 P5, L3: What is 'localization and imbalance'?

6 P5, L26: Please define $Y\_k$, or is it the same as $Y\_{k,t}$.

7 P7, L3-5: Move this paragraph to the end of section 2.2. I was actually wondering whether twenty cases were sufficient to compute statistics.

8 P7, L30: How are model years defined?

9 P8, L9-11: What do you mean by the strong nonlinearity and smoothness of the Runge-Kutta method? Can you elaborate on this point? Perhaps a different method

would be more appropriate to obtain the solution of system (1).

10 P8, L22: It is not clear why the spread should be small. Please explain this point.

11 P8, L28: What is 'convexity with respect to OCN-OTWs'?

12 P11, L1: Which one is the curve of X2-RMSE in the OCN-OTW space? The figure (Fig. 9b) only shows one set of lines as a function of OCN-OTW and it corresponds to omega-RMSE.

13 P11, L12-17: The information in these lines is a repetition from the previous paragraphs. They can be deleted.

14 P12, L3-8: The discussion about the influence of the coupling term in the optimal OTWs is interesting, but it is hard to see what the implications for the real world or for a realistic CDA system are. Please, also discuss these points.

15 Figure 5: What does the blue shading under the lower bound mean?

16 Figure 10c is not discussed and therefore could be dismissed.

17 Figures 11a and specially Fig. 11b, which only shows a horizontal line, are not needed as they can be simply described in the text.

Technical corrections:

P1, L21-23: Some of the main results are given in these lines. However, the sentence is written in a very confusing style. I would encourage the authors to rewrite it to make the abstract, and the paper, more accessible.

P1, L30: Delete 'the' between 'by' and 'coupled'.

P1, L32: Add 'to sub-grid processes' after 'approximation'.

P2, L11: Delete 'in each medium here' and add 'for each medium' after '(OTW)'.

P2, L20: Delete 'exist' between 'not' and 'an' and add 'exists' after 'OTW'.

P2, L21: Expand 'What's' to 'What is'.

P2, L24: Change 'identify' to 'identifies'. Also, the end of the sentence, after 'medium' is not clear. Please, rewrite.

P3, L25: Change 'frequent' to 'frequency'.

P4, L7: Delete 'Namely'.

P4, L8: Define TU.

P4, L12: I think the sentence is incomplete. What does 'C2' does in contrast?

P4, L16-17: As it is written, the sentence starting with 'Where' and ending with 'scale' makes no sense. Please, rewrite.

P4, L21: Do you mean 'In summary' rather than 'Summarily'?

P4, L24: The method's name is Runge-Kutta not Runger-Kutta. Since it is not used in the text the acronym RK4 is not necessary.

P4, L25: Change notation as 'k0-k3' looks like 'k0 minus k3'.

P5, L6: Either change 'In the words of Zhang (2011a)' to 'Following Zhang (2011a)' or clearly indicate the quote from that paper using quotation marks.

P5, L14: The sentence does not make sense. Please rewrite.

P6, L15: Change 'including' to 'included'.

P6, L19 and L20: Change 'integrates' to 'is integrated' P6, L24: Delete 'etc.' or perhaps change it to 'and other parameters'. P7, L4: Change 'will obtain' to 'was obtained'. P7, L11: Omega and w are two different letters. P7, L28: The sentence is incomplete. Figure 3 presents the power spectrum of what? P8, L23: Change 'keeps' to 'is kept'. P9, L26 and L27: Change 'including' to 'included'. P10, L15: Delete 'gained'. P10, L11: I think where it says 'only limits' should say 'is limited'. Caption to Fig. 9: Change 'panels' to 'panel'. Figure 10: The value of C1 in the legend is constant and therefore
it is not needed there.

---

## Author Comment (AC1) · 21 Apr 2017

Reviewer 1# A few tel-conferences of all co-authors have been held to discuss the comments from reviewer #1. All authors converge to the point that all the comments are very important and useful for authors to improve the quality of this manuscript (MS). Therefore, all comments from reviewer #1 have been fully addressed in the revision.

Now we will reply to each comment point by point as following:

1 the title does not fit. You are not discussing the "Impact of Optimal Observational Time Window' but "Impact of Observational Time Window". It may be "Impact of Observational Time Window in Coupled Data Assimilation with a Simple Climate Model".

RE: Thank you very much for your generous comment and we fully agree with that. We have corrected the tile as "Impact of Observational Time Window on Coupled Data Assimilation: Simulation with a Simple Climate Model".

2 The concept of OTW in this study validates the observations in a time window to the center of the window (analysis time). It is very useful technique for data assimilation (DA). More or less it has been applied in the assimilation with real observations. Here the OTW in this study is applied in 3-dimension DA but not 4-dimension DA, which need address in the introduction. The citations of OTW are not very relevant (Page 2, line 13-15). I could not find the clear concept of OTW and how they applied in data assimilation. So you should address the technique details in the paper. I guess that you treat all the observations independently and assimilate then sequentially using their original error scales. another approach is to assimilate the average values of the observations in the OTW. It is worth to compare these two approach in the study.

RE: As Line 19-23 of Page 2, we have rewritten this part and addressed that in this study the OTW validates the observations including in a time window to the center of the window and are sequentially assimilated using their original error scales. The irrelevant citations have been deleted and we have added some ideas in a relevant one to introduce the concept of OTW using in this study. As Line 8-18 of Page 12 in the revised manuscript, we have added some discussion about the comparison of sequential assimilation and averaged-observational assimilation approaches. And the comparison of the results of the two methods are shown as Figure. 1. Here the Sqe-CDA-OTW represents the experiment that sequentially assimilates the observations including in the OTWs using their original error scales. And the Ave-CDA-OTW denotes the experiment that assimilates the average of the observations including in the OTWs. But the standard deviation of the observational error will be $1/(\sqrt{N})$ of the original error scales. Here all the experiments will use the biased model setting and the uni-variate adjustment scheme. From Fig. 1, we can see that the results of the Ave-CDA-OTW is almost same as those of the Sqe-CDA-OTW. And we will illustrate

this phenomenon form the perspective of the calculation of the EAKF method (the corresponding interpretation are shown following Fig.1). And we can see that for a CDA system in the CGCM with a suitable inflation scheme, the assimilation approach with the time-averaged observations will have more advantages over that which sequentially assimilate the observations. It is very interesting to comparing these two approaches, but using either assimilation approach will have no influence on the essential of this study. Thanks.

3 From your result I can not connect the de-correlation time with the optimal OTW directly. The optimal OTW is much short than the de-correlation time. Like in figure7, the correlation is 0.995 for the optimal OTW. Please revise the abstract and conclusion. The time scales of the variables really can not provide the useful information to quantify the optimal OTW. I suggest you remove or shorten the discussion and analysis related to the time scales of the variables.

RE: As Line 19-27 of Page 10, We have shortened this part. On the one hand, we have deleted the discussion about the relationship between the length of the optimal OTWs and the de-correlation time scales. On the other hand, we reserved the discussion about the high correlation between the observations including in the optimal OTW and the model state at the analysis time, which makes sense to the main idea of this study. Thanks.

4 The time (s) of observations in the OTW are different from the assimilation time. The temporal offsets introduce the represent errors for the observations, which need to be consider during analysis. If not, EnKF overweights the observations The analysis will not be optimal and the analysis ensemble spread will has negative biased. Please address this part and justify your assimilation.

RE: This is a very important and necessary comment. As Line 19-33 of Page 12, we have added some discussion about the temporal offset introduced the representation errors for the observations that are different from the analysis time. We use the decorrelation coefficients to weight the observations including in the OTWs and avoid the overweighting. The comparison of the results of the weighted and no-weighted experiment are as Fig. 2. We can see that the lengths of the optimal OTWs obtained by these two assimilation schemes are the same except that the RMSEs of the weighted observation experiment will be lower than that of the non-weighted observation experiment when using the longer OTWs (when the length of ATM-OTW is greater than 4 and that of the OCN-OTW is larger than 50). This is owing to the high correlation between the observation including in the optimal OTWs and models states at the analysis time (exceed 0.995). Thus the influence of the temporal offset can be ignored and the results obtained by these two scheme will be almost same when using the lower OTWs (no or less greater than the lengths of the optimal OTWs). When we use the longer ones, the correlation will decrease and influence of the temporal offsets will be obvious that the results of the weighted observation experiment will better. For the CGCMs, owing to the complex physics and dynamics, the influence of the time offsets will be obvious and the weights of the observations will be very necessary. But from this simple model case, we can see that whether or not using the weighted observations, the relationship between the characteristic variability time scales and the optimal OTWs will be firm and the essence of this study is established. Thanks.

5 It is hard to find the assimilation intervals for different components. Please address them together in experiment setup (2.3). I noticed that the OTW is longer than the assimilation interval, which means that you assimilate the same observations multiple times. In the perfect framework, it should degrade the filter performance from my understand and the information theory. Can you explain why you can achieve a good analysis under that circumstance. For a biased model framework, people do use observation multiple times to compensate the negative bias of forecast ensemble spread. So it make sense to see the optimal OTW in biased model cases are longer than that in perfect model case.

RE: As Line 21-22 of Page 7, in this study the assimilation interval of the atmospheric

and slab oceanic states are 5 and 20 time steps, respectively. But the optimal OTWs obtained for the atmospheric and oceanic component in the perfect experiment framework are 1 and 10, which shows that no observation will be assimilated for multiple times. When the length of the OTW increases beyond that of the optimal one, the filter performance degrades, which satisfy your understanding and the information theory. Thanks.

6 The section 2.2 of "Ensemble coupled data assimilation" is incomplete. People will not understand your equations without trace back to your references. Please provide the completed two steps of EAKF.

RE: This comment is necessary. As in the section 2.2 in the revised manuscript we have provided the completely two steps of EAKF. Thanks.

7 Page 9 the last paragraph. The discussion of OTW with 4D-var is very confusion and hand-waving. The OTW in 4D-var is very different concept than yours. You should remove i

RE: Have removed it. Thanks.

8 Figure 4(a & b) show that the lower bound of RMSE reaches to 0. That does not make sense. Please check.

RE: Figures 4 and 7 are re-plotted. Thanks.

9 Figure 4 and Figure 5 show that the ensemble spread is significantly smaller than the RMSE. Please address the reasons and the effect on the assimilation.

RE: As Lines 14-18 of Page 10 of the revised manuscript, is this study, no inflation scheme was used in all the assimilation experiments. And only the data obtained in the last 5000 TUs will be used to conduct the error statistics for evaluation. Thus after first 5000 TUs assimilation, the ensemble spread has been greatly reduced and is significantly smaller the corresponding RMSE. Although in theory the reduced ensemble spread will degrade the quality of state estimation, in this study with a one-dimensional

conceptual coupled model, for simplicity and computational convenience as well as convenience for comparison, we insist not using the inflation scheme. Thanks.

10 P11 Lin13 I can not buy the statement of "since more observational information is needed to compensate the model bias and recover the characteristic variability." The reason should be "the forecast ensemble in a biased model underestimates the forecast error, which results the EnKF underweight the observations. Therefore one can improve filter performance by using observation multiple times"

RE: We fully agree with your point. In the biased model experiment framework, the length of the optimal OTWs is larger than those in the perfect experiment case. The only reason will be the influence of the bias in the biased model. As Line 21-27 of Page 11, we think that we need stronger observational constraints to reduce the mean discrepancy between the model and observation induced by the influence of the bias. Also we fully agree that it needs to assimilate the observations for multiple times to compensate the underestimation of the forecast error by the bias in the biased model. We think these two interpretations are identical that using larger OTWs to compensate the influence of the bias except from two different aspects. Thus we think these two interpretations are not contradictory. Here we reserve our original interpretation and add your point into the revised manuscript. Thanks.

11 A reading proof is required to improve the manuscript. There are too many typos and grammar errors. For example Page 2 Lin 33 The "ensemble filter" should be "ensemble Kalman filter" Page 6 Lin 13 "but they are started from different initial states." can be deleted Page 6 Lin 15 "observation" should be "truth" Page 6 Lin 33 "the coupling co-efficient C1is sensitive to model stability" should be "the the model stability is sensitive the coupling coefficient C1" Page 7 Lin3-5 The whole sentence need rephrase. You can just mention that "Following Zhang and Anderson (2003) an ensemble size of 20 is applied in all experiments in this study." Page 7 Lin9-11 "absolute error" is not un-derstandable. I guess that is "the absolution values of error". The experiments are not described clearly. Please rewrite. Page 7 Lin28 the sentence is not completed. Page

7 Lin 30 Please also give the equivalent time of atmosphere (1-2 model month?). The deep ocean should be 50 model year.

RE: Thank you very much for your indication. All of these are fixed. Here the "absolute error" just represents the "absolute value of the difference between the estimated model state value and corresponding truth". Second in the section 2.3 of the revision manuscript, the experiments have been rewritten and described clearly. As Line 28-29 of Page 8, the equivalent time of atmosphere is 1-2 model month and that of the deep ocean should be 5 model decades.

Thank you very much again for your generous help and comments! We hope that the revised manuscript meets your requirement.

Sincerely yours, Shaoqing Zhang and Co-author

Please also note the supplement to this comment:
http://www.nonlin-processes-geophys-discuss.net/npg-2016-68/npg-2016-68-AC1-supplement.pdf
* * *
[Figure]

[Figure]

Figure. 1: The results of the comparison of sequential assimilation and averaged- observational assimilation approaches.

Here we assume the OTW including two observation $a$ and $b$,

1) Sequentially assimilate these two observations using their original error scales:

$$\Delta Z_{i,j} = \frac{Cov(Z_j, Y_k)}{(\sigma_{k,k}^p)^2}\left[\left(\frac{\bar{Y}_k^p}{1+r_k^p} + \frac{Y_k^o}{1+r_k^{-z}} + \frac{\Delta Y_{k,i}^p}{\sqrt{1+r_k^z}}\right) - Y_{k,i}^p\right]$$

in this study there is no inflation scheme, thus after many assimilation steps, the ensemble spreads of the model states will greatly decreased will be greatly smaller than the observational error. Thus we can assume $\bar{Y}_k^p \approx Y_{k,i}^p$ and $\frac{Cov(Z_j, Y_k)}{(\sigma_{k,k}^p)^2} \approx constant\ r$. For simplicity and convenience, we assume at a assimilation time the $\bar{Y}_k^p$, $\sigma_{k,k}^p$ and $\Delta Y_{k,i}^p$ will keep as constants during the sequential data assimilation process. And we can use constants $C_1$, $C_2$ and $C_3$ to represent the $\bar{Y}_k^p$, $r_k^2$ and $\Delta Y_{k,i}^p$, respectively.

$$Z_{i,j} = \frac{1+(1-r)C_2}{1+C_2}\left[\frac{1+(1-r)C_2}{1+C_2}C_1 + \frac{rC_2}{1+C_2}a + \frac{r}{\sqrt{1+C_2}}C_3\right] + \frac{rC_2}{1+C_2}b + \frac{r}{\sqrt{1+C_2}}C_3$$

$C_2$ is very small and we can assume that $C_2{}^2 \approx 0$. Thus

$$Z_{i,j} \approx \frac{1+2(1-r)C_2}{1+2C_2}C_1 + \frac{rC_2}{1+2C_2}(a+b) + \frac{2r+2rC_2-r^2C_2}{\sqrt{(1+C_2)^3}}C_3$$

2) assimilate the average of the observations but $C_2{}' = (r_k{}')^2 = 2C_2$; thus

$$Z_{i,j}{}' = \frac{1+2(1-r)C_2}{1+2C_2}C_1 + \frac{2rC_2}{1+2C_2}\frac{(a+b)}{2} + \frac{r}{\sqrt{1+2C_2}}C_3$$

And we can see that the difference will be the third term $\frac{2r+2rC_2-r^2C_2}{\sqrt{(1+C_2)^3}}C_3$ and $\frac{r}{\sqrt{1+2C_2}}C_3$.

Method 1: $\frac{2r+2rC_2-r^2C_2}{\sqrt{(1+C_2)^3}}C_3 \approx \frac{2r+2rC_2-r^2C_2}{1+C_2}C_3$

Method 2: $\frac{r}{\sqrt{1+2C_2}}C_3 \approx \frac{r}{1+C_2}C_3$

Method 1-Method 2, thus:

$$\frac{2r+2rC_2-r^2C_2}{1+C_2}C_3 - \frac{r}{1+C_2}C_3 = \frac{r+2rC_2-r^2C_2}{1+C_2}C_3$$
$$= \frac{-C_2(r-1)^2 + r + C_2}{1+C_2}C_3 > 0\ and\ < C_3$$

Here $C_2$ and $C_3$ are also very small, thus the results of these two methods are almost same. And we can extend that to longer OTWs and more observations cases in this study.

**Fig. 1.** The results of the comparison of sequential assimilation and averaged- observational assimilation approaches.

We use the de-correlation coefficients (As the Panel a and b of the following figure) to weight the observations including in the OTWs and avoid the overweighting. The comparison of the results of the weighted and no-weighted experiment are as following:

[Figure]

Figure 2: The comparison of the results of the weighted and no-weighted experiment.

**Fig. 2.** The comparison of the results of the weighted and no-weighted experiment.

---

## Author Comment (AC2) · 21 Apr 2017

Dear O. Martinez Alvarado,

A few tel-conferences of all co-authors have been held to discuss yours comments. All authors converge to the point that all the comments are very important and useful for authors to improve the quality of this manuscript (MS). Therefore, all comments of yours have been fully addressed in the revision.

Now we will reply to each comment point by point as following:

1 It would be useful to have more background information on the CDA strategy studied in the paper. In particular I was wondering why the observational data should be

assumed at the assimilation time when 4DVar would avoid that problem. It was only in Section 4 that 4DVar was mentioned as an alternative.

RE: As Line 10-16 of Page 2, we have added some background information on the CDA strategy in the introduction part of the revised manuscript. A 4-dimensional variation (4D-Var) scheme implements state estimation through minimizing a distance measure between the model and observations defined on all individual observational times within an OTW. Thus the minimized distance reflects the averaged effect in the OTW. As in Hunt et al. (2004), we expand the EnKF to include a time window in which the observations are treated as the exact assimilation times, even though their times are different in the window. Namely, we just assume that all the collected data sample the "truth" variation at the assimilation time and will be sequentially assimilated with their original error scales. Thus in this study the observational data included in the OTWs should be assumed at the assimilation time while 4DVar can relax that problem. Thanks.

2 P2, L20-22: The two questions posed are circular in the sense that the answer to the first would depend on the effects of varying the OTW on the quality of CDA and the answer to the second depends on the existence of an optimal OTW. I would suggest the following rearrangement: '1) What is the impact of varying OTWs for each coupled component within the coupled model framework on the quality of CDF for climate estimation and prediction initialization? 2) Based on this impact, Is there an optimal OTW so that assimilation fitting has maximum observational information, but minimum variability distortion?' The point is subtle, but it might be worth doing for clarity.

RE: Thank you very much for your generous comment and we fully agree with that. We have corrected that as Line 27-29 of Page 2 in the revised manuscript.

3 Equation (1): Should there be a dot over eta? Is the dot over C6 correct?

RE: Fixed. Thanks.

4 It is not necessary to describe the Runge-Kutta method in Section 2.1.

RE: As Line 3-4 of Page 5, we have deleted the discussion and equation of the Runge-Kutta method. Thanks.

5 P5, L3: What is 'localization and imbalance'?

RE: Because of the sampling error from a finite ensemble size, the ensemble-evaluated background variance is usually underestimated, and spurious correlations exist between a state variable and remote observations. To remove the long-distance spurious correlations and increase the reliability of ensemble-evaluated background covariance, the localization technique was introduced into ensemble-based filters. Given a biased CGCMs and the atmospheric and oceanic observing system (also including the land and sea ice), maintaining a balanced and coherent climate estimation is of critical importance for producing accurate climate analysis and prediction initialization. Given the assimilation model bias of warmer atmosphere and colder ocean, the atmospheric-only (oceanic only) data constraint produces an overcooling (overwarming) ocean through the atmosphere–ocean interaction will create imbalanced and incoherent oceanic (atmospheric) states estimation contrast to the observational model. The imbalanced atmosphere-to-ocean flux tends to decrease the assimilation quality and produce different climate features and variability from the real world. But this one-diamond conceptual coupled model is too simple to simulate the complex physics such as the imbalance between different components and have no need to apply the localization scheme and consider the imbalance.

6 P5, L26: Please define $Y_k$, or is it the same as $Y_{k,t}$.

RE: In this study the $y_k$, or is it the same as $y_{k,t}$, as Line 24 of Page 5. Thanks.

7 P7, L3-5: Move this paragraph to the end of section 2.2. I was actually wondering whether twenty cases were sufficient to compute statistics.

RE: This simple conceptual coupled model is same as that used in Zhang et al. 2012

(A study of enhancive parameter correction with coupled data assimilation for climate estimation and prediction using a simple coupled model). In that study, based on the trade-off between cost and assimilation quality, after a series of sensitivity tests on ensemble sizes of 10, 20, 50 and 100, no significant difference on assimilation quality is found when the ensemble size is greater than 20. Thus we chose the ensemble size of 20 in this study. And in order to avoid causing confusion, we correct that as Line 28-29 of Page 7 of the revised manuscript. Thanks.

8 P7, L30: How are model years defined?

RE: A model year is defined as 10 non-dimensional time units, as Line 24 of Page 6. Thanks.

9 P8, L9-11: What do you mean by the strong nonlinearity and smoothness of the Runge-Kutta method? Can you elaborate on this point? Perhaps a different method would be more appropriate to obtain the solution of system (1).

RE: Usually the Leap-frog method and Rugge-Kutta method will be used as the time difference time scheme to obtain the solution of the climate models. In the paper named "Mitigation of coupled model biases induced by dynamical core misfitting through parameter optimization: simulation with a simple pycnocline prediction model", Fig.3 showed the power spectra of this simple coupled model's states derived by the Leap-frog and RK4 time difference scheme. We can see that the characteristic variability time scale of each coupled components obtained by both methods is a little larger than that set in Eq. (3) and varies for different time difference time scheme selected. The main idea of this study just illustrates that a closely relationship between the length of optimal OTWs and the corresponding characteristic variability time scales exists, which is independent from the time difference scheme we selected in this study. Owing to the simplicity and directness, we choose the RK4 as the time difference scheme used in this study. But its strongly nonlinearity and smoothness of the Runge-Kutta method will smooth the solution of the time series of the solutions, the characteristic variability time

scale will be extended and larger than that set originally. Thanks.

10 P8, L22: It is not clear why the spread should be small. Please explain this point.

RE: In this study each assimilation experiment will be repeated for 20 times starting from 20 different ensemble initial conditions. The mean value and uncertainty (standard deviation) of the 20 cases will be evaluated. Here the spread just represents the standard deviation of the 20 cases. Thus the spread will be smallest when using the optimal OTW, as Line 17-19 of Page 9. Thanks.

11 P8, L28: What is 'convexity with respect to OCN-OTWs'?

RE: Sorry for this mistake and confusion. And the "convexity" has been replaced by "sensitive variation" which just represents the variation of the Omega-RMSE curves is sensitive with respect to the OTWs. As Line 25 in Page 9 of the revised manuscript we have corrected. Thanks.

12 P11, L1: Which one is the curve of X2-RMSE in the OCN-OTW space? The figure (Fig. 9b) only shows one set of lines as a function of OCN-OTW and it corresponds to omega-RMSE.

RE: In this study the curve of X2-RMSE in the OCN-OTW space is not shown and the reason has been added as Line 25- 27 in Page 9 in the revised manuscript. Thanks.

13 P11, L12-17: The information in these lines is a repetition from the previous paragraphs. They can be deleted.

RE: Corrected. Thanks.

14 P12, L3-8: The discussion about the influence of the coupling term in the optimal OTWs is interesting, but it is hard to see what the implications for the real world or for a realistic CDA system are. Please, also discuss these points.

RE: As we have shown that changing that changing coupling strength (controlled by the coupling coefficients C1 and C2 in this case) between the atmosphere and upper

ocean may have some influence on the characteristic variability time scales of coupled media, so as on the optimal OTWs. On the one hand these experiments can further illustrate the idea that a close relationship between the length of the optimal OTW and the corresponding characteristic variability time scale exists. On the other hand, for a realistic CDA system, the coupling physics could be very complicate and affected by many factors. The results of this simple model give the insights that we can only consider the factors which have obvious influence on the characteristic variability time scales when determining the length of the optimal OTWs for a realistic CDA system. And the factors which will have litter influence on the characteristic variability time scales can be just ignored. Through this way the process of determining the optimal OTWs for the realistic CDA system will be greatly simplified and make it possible to apply the method of using the optimal OTWs to the realistic CDA system. As Line 22-28 of Page 13 and Line 20-23 of Page 14, we have added some discussion on these points. Thanks.

15 Figure 5: What does the blue shading under the lower bound mean?

RE: As Line 8-9 of Page 10 and Line 3-4 of Page 20 of the revised manuscript, the blue shading below the ensemble spread represents the range of the uncertainty of state estimation in each assimilation experiments. Thanks.

16 Figure 10c is not discussed and therefore could be dismissed.

RE: From panel c) we can clearly see that the time series of the Omega also obviously varies with changing C2, which also corresponds to the results shown in panel b) in Figure 10. Thus we think that panel c) is also necessary and reserve it. Thanks.

17 Figures 11a and specially Fig. 11b, which only shows a horizontal line, are not needed as they can be simply described in the text.

RE: We fully agree with this comment and in the revised manuscript the panel a) and b) of the Figure 11 have been deleted. Thanks.

Technical corrections: P1, L21-23: Some of the main results are given in these lines. However, the sentence is written in a very confusing style. I would encourage the authors to rewrite it to make the abstract, and the paper, more accessible.

RE: This sentence has been rewritten, as Line 21-22 of Page 1. Thanks.

P1, L30: Delete 'the' between 'by' and 'coupled'. P1, L32: Add 'to sub-grid processes' after 'approximation'. P2, L11: Delete 'in each medium here' and add 'for each medium' after '(OTW)'. P2, L20: Delete 'exist' between 'not' and 'an' and add 'exists' after 'OTW'. P2, L21: Expand 'What's' to 'What is'. P3, L25: Change 'frequent' to 'frequency'. P4, L7: Delete 'Namely'. P5, L6: Either change 'In the words of Zhang (2011a)' to 'Following Zhang (2011a)' or clearly indicate the quote from that paper using quotation marks. P6, L15: Change 'including' to 'included'. P6, L19 and L20: Change 'integrates' to 'is integrated' P6, L24: Delete 'etc.' or perhaps change it to 'and other parameters'. P7, L4: Change 'will obtain' to 'was obtained'. P7,L11: Omega and w are two different letters. Figure 3 presents the power spectrum of what? P8, L23: Change 'keeps' to 'is kept'. P9, L26 and L27: Change 'including' to 'included'. P10, L15: Delete 'gained'. P10,L11: I think where it says 'only limits' should say 'is limited'. Caption to Fig. 9: Change 'panels' to 'panel'.

RE: Corrected. Thanks.

P2, L24: Change 'identify' to 'identifies'. Also, the end of the sentence, after 'medium' is not clear. Please, rewrite.

RE: Corrected. (The sentence has been corrected as Line 30-31 of Page 2 in the revised manuscript). Thanks

P4, L8: Define TU.

RE: As Line 9-11 0f Page 4, TU is the time unit. And in this study 1TU equals 100 time steps. Thanks.

P4, L12: I think the sentence is incomplete. What does 'C2' does in contrast?

RE: In this study C1 denotes the upper slab oceanic forcing on the atmosphere while C2 in contrast. Here 'C2 in contrast' just represents that C2 denotes the atmospheric forcing on the upper slab oceanic, as Line 19 of Page 4. Thanks

P4, L16-17: As it is written, the sentence starting with 'Where' and ending with 'scale' makes no sense. Please, rewrite.

RE: This sentence has been rewritten as Line 24-26 of Page 4 in the revised manuscript. Thanks.

P4, L21: Do you mean 'In summary' rather than 'Summarily'?

RE: Yes. Corrected. Thanks.

P4, L24: The method's name is Runge-Kutta not Runger-Kutta. Since it is not used in the text the acronym RK4 is not necessary.

RE: In the revised manuscript we have corrected as 'Runge-Kutta' and deleted the acronym RK4. Thanks.

P4, L25: Change notation as 'k0-k3' looks like 'k0 minus k3'.

RE: According to a reviewer's generous comment, we have deleted the discussion about the RK4. Thanks.

P5, L14: The sentence does not make sense. Please rewrite.

RE: This sentence has been rewritten as Line 18-22 of Page 5 of the revised manuscript. Thanks.

P7, L28: The sentence is incomplete. Figure 3 presents the power spectrum of what?

RE: Corrected as Line 26 of Page 8 of the revised manuscript. Thanks.

Figure 10: The value of C1 in the legend is constant and therefore it is not needed there.

RE: We have deleted the C1 in the legend in Figure 10. Thanks.

Thank you very much again for your generous help and comments! We hope that the revised manuscript meets your requirement.

Sincerely yours, Shaoqing Zhang and Co-author

Please also note the supplement to this comment: http://www.nonlin-processes-geophys-discuss.net/npg-2016-68/npg-2016-68-AC2-supplement.pdf

**Supplement:**

[revised manuscript text omitted]

---

## Author Response (AR2)

Reviewer #2 Oscar Martinez-Alvarado:

Thank you very much for your generous suggestions and comments, which are very important and useful for authors to improve the quality of this manuscript (MS). Therefore, all comments of yours have been fully addressed in the revision. And we have run through this manuscript for many times to make sure that the presentation of the English is up to standards and the manuscript is easy to read.

Now we will reply to each comment point by point as following:

I have fully read the new version and I found that there are several typos and minor errors that should be corrected before the article is published in Nonlinear Processes in Geophysics. I include a list of comments. However, this might not be exhaustive. I believe the most appropriate course of action would be to run this through an expert in the English language:

In general: There are several instances where a present participle is used instead of a past participle and vice versa (e.g. including instead of included, resulted instead of resulting).

RE: Thank you very much for your generous comment and we fully agree with that. As Line 22 of Page 1 and Line 16 of Page 12, the "resulted" has been corrected as "resulting". As Line 4 of Page 2, Line 5, 29 of Page 4, Line 24, 26 of Page 10, Line 26 of Page 11, Line 8, 10, 11, 21 of page 12, the "including" has been corrected as "included".

P2 L9: Change 'improve' for 'improving'.

P2 L13: Change 'theoretical and technical challenges' for 'theoretically and technically challenging'.

P4 L4: you can include '\omega' after 'slab oceanic variable' for clarity.

P4 L15: Delete 'And' at the beginning of the sentence.

P4 L16: Use the plural of parameter.

P4 L24: Change '... \Gamma keeps as a constant…' for '... \Gamma is kept constant…'

P7 L14: Delete 'for' after 'repeated'.

P7 L21: Delete period after 'Figure' in 'Figure. 1'.

P8 L9: Delete 'And' at the beginning of the sentence.

P8 L10: Do not capitalise 'NOT'.

P9 L29: Change 'expended' for 'expanded'

RE: Corrected as above. Thanks.

P10 L11: The sentence beginning with 'While increasing…' does not make sense. Please, rewrite it.

RE: This is a very important and necessary comment. We have rewritten this sentence as Line 11-14 of Page 10, which makes sense.

P10 L22: Delete brackets around 'steps'.

P10 L26: Change 'arrange' for 'range'.

P12 L12: Change 'is' for 'are'.

P12 L8-33: There are several places where 'same' should be change to 'the same'.

RE: Corrected as above. Thanks.

P12 L18 and 32: I'm not sure that the word 'firm' is being used correctly.

RE: Thank you very much for your generous comments. In Line 18 of Page 12, we think that "firms" is suitable. And in Line 32 we have corrected "firm" as "robust".

P12 L28: Delete 'And' at the beginning of the sentence.

P13 L2: Insert 'the' between 'Changing' and 'coupling'.

RE: Corrected as above. Thanks.

Thank you very much again for your generous help and comments!

We hope that the revised manuscript meets your requirement.

Sincerely yours,

Shaoqing Zhang and Co-author

[revised manuscript text omitted]